# Modeling of the Effect of Subperiosteal Hydrostatic Pressure Conductivity between Joints on Decreasing Contact Loads on Cartilage and of the Effect of Myofascial Relief in Treating Trigger Points: The Floating Skeleton Theory

**DOI:** 10.3390/biomimetics9040222

**Published:** 2024-04-08

**Authors:** Mark R. Pitkin

**Affiliations:** 1Department of Orthopaedics & Rehabilitation Medicine, Tufts University School of Medicine, Boston, MA 02111, USA; mpitkin@tuftsmedicalcenter.org; Tel.: +1-339-364-1955; 2Poly-Orth International, Sharon, MA 02067, USA

**Keywords:** cartilage overloading, subperiosteal pressure transmission, floating skeleton concept

## Abstract

Chronic overloading of the cartilage can lead to its irreversible destruction, as observed in people with osteoarthritis. The floating skeleton model previously introduced postulates that overloading begins and progresses when a joint is isolated from the hydrostatical connection with other joints. Such a connection occurs via the interstitial fluid in subperiosteal space and allows for pressure transmission between synovial capsules modulating intra-articular pressure. In the current study, a simple experiment was performed to model an obstruction in the subperiosteal hydrostatic pressure conductivity between joints to illustrate the effect of that obstruction on loads borne by the joint. When the obstruction was removed, the load experienced by the joint was reduced as it was redistributed throughout the model structure. The experiment demonstrated that contact pressures can be redistributed when the conditions of Pascal’s Law are met.

## 1. Introduction

According to computational modeling, the cartilage in sound joints during locomotion is subject to crushing loads. For example, a model of contact pressure in sound knee joints during regular gait has shown a peak pressure during stance of 6.5 MPa [1]. Perplexingly, such contact pressure levels are higher than cartilage, structurally, should be consistently able to bear [2]. Moreover, the effect is not confined to theoretical computations; the in vivo measurement of maximum contact pressures on acetabular cartilage from implanted, instrumented femoral head hemiprostheses are likewise 5–6 MPa in gait and even 18 MPa with more aggressive movements [3]. By way of comparison, the pressure exerted by a tank on the ground is about 0.55 MPa, while the pressure under stilettos is about 13 MPa [4].

Over two dozen theories for the extraordinary performance of synovial joints have been put forth to attempt to explain their high load capacity, but as is usually the case when so many theories abound, the mechanisms involved are far from being understood [3]. Multiple studies demonstrate the unique morphology of contacting cartilage, including specific reactions to the indentation of the cartilage tissue [5,6], the exceptional lubrication capacity of the synovial fluid, with data on the role of its composition [7], and pressurization ability [8]. In all of these studies, the joint capsule is presented as a closed system separated from the other joint capsules. We theorize that the capsules may, in fact, be connected as an articular system through microscopic channels impregnated with interstitial fluid running between joint capsules in a tiny space between the periosteum and the bone.

In 2015, the authors’ group published an experimental animal study to support this theory [9]. Contradicting traditional physiology and anatomy [10] and in contrast with previous studies reporting changes in intra-articular pressure in a passively ranging single joint [11,12], we were the first to report on the pressure changes in uninvolved joints when one distant joint was passively ranged. 

As for single joint articulation, when changing the curvature of the synovial capsule, the change in the intra-joint pressure is explained by the Laplace Law, stating that the pressure of a gas or fluid inside an elastic container with a curved surface is inversely proportional to the radius as long as the surface tension is presumed to change little [13]. This explains the results of a report by Levick [11] on the increase in synovial fluid pressure in a rabbit knee due to acute flexion, and Tarasevicius et al.’s report on the mean hydrostatic intracapsular pressure increase in a human hip at 45° of flexion [14], as well as the intermediate results of our study [9].

The findings suggested a physiological mechanism through which pressures are not entirely contained within joint capsules (Figure 1a), but are “shared” between joints (Figure 1c). A critical hydromechanics consequence is that the actual loads borne by cartilage are lower than what is mathematically computed for the joint capsule, since the computation does not account for the sharing of pressure among the capsules. The findings more straightforwardly explain how, normally, cartilage can remain healthy for decades. The findings also suggest that when pressure sharing is disrupted, the pressure borne by a given cartilage spikes; this logically leads to negative consequences to its health and integrity.

In a study on 10 rabbits [9], we considered pairs of joints, one of which was passively articulated while the other was immobilized, and simultaneously measured the pressure changes inside each (Figure 1b; Appendix A). The first research question was whether pressure changes in an articulated joint cause pressure changes in the paired, uninvolved stationary joints, and if they do, whether the uninvolved joint has to be adjacent to the articulated joint or if it can be distant.

The answer was yes: when the rabbit’s joint was passively ranged, statistically significant changes in intra-articular pressure inside the paired, uninvolved joint were detected. The result held true for all joint pairs that were considered, both when they were adjacent (e.g., ipsilateral ankle and knee) and when they were distant (e.g., contralateral knees). 

The second research question was whether there is evidence that the pressure is transmitted subperiosteally. The experiment was repeated after transecting the periosteum above the ranged joint to create a condition in which pressure cannot be transmitted subperiosteally. Crucially, after the periosteal transection, while pressure changed inside the ranged joint in response to articulation, pressure in the uninvolved, stationary joints remained constant Appendix A. Perhaps pressure from the capsule hydrostatically propagates through the fibrous stratum in the groove of Ranvier [15]. A detailed morphological and physiological description of this system will be the subject of future multidisciplinary studies.

Because these physiological results were not foreseen by any preexisting theory, what is required now, and what is presented below in the Methods section, is a hydromechanical, experimental demonstration of how loads applied to contact cartilage with an interrupted interarticular fluid transmission (experiment 1) are reduced when such transmission is restored (experiment 2). We built a simple hydromechanical model of the subperiosteal space, whose characteristics meet the conditions of Pascal’s Law, stating that pressure applied to an enclosed fluid will be transmitted without a change in magnitude to every point of the fluid and to the walls of the container even without the fluid flowing [16]. 

We believe that this demonstration can enhance our understanding of the musculoskeletal hydromechanical mechanism, which may be also used for a more effective practice of various rehabilitative exercising methods such as the Sanomechanics approach [15,17]. 

## 2. Methods

The morphology of the periosteum, which lines the outer surface of all bones except the sesamoid bones and the intra-articular ends of bones [18,19], supports the possibility of hydrostatic conductance between capsules of adjacent and distant joints through channels between the bone and the periosteum. It is sufficiently flexible: the modulus of elasticity in various parts of the periosteum ranges between 63.0 ± 25.4 MPa and 91.7 ± 30.5 MPa [20] (compared to 0.4 MPa for venous capillaries [21]). It is also non-stretchable, as the superficial portion of the outer layer is formed by a collagenous matrix consisting of small compact bundles with interspersed elongated fibroblasts [19]. This explains the ability of the periosteum to withstand substantial tensile strains [22] and makes it eligible to support the hydrostatic transmission of pressures between joint capsules via the subperiosteal fluid in the microscopic space between the periosteum and the bone’s outer surface [9]. 

### Hydromechanical Model

To model the obstructed versus unobstructed subperiosteal hydrostatic transmission of pressures, we used a 16.5 cm × 14.9 cm Ziploc™ bag made of low-density non-stretchable and flexible polyethylene (LDPE). A teaspoon of melted sugar was placed inside the bag, roughly in its center. After cooling, the sticky sugar caramelized into a circular plug, firmly gluing together the walls of the bag. The bag with the plug was filled with 75 mL of water (Figure 2(1A)). Adding a water layer between the bag’s walls effectively transformed it into a cushioning pad able to hydrostatically distribute local pressures. At the same time, the caramel plug excluded a small zone defined by the plug’s diameter from pressure transmission. 

This construct simulated the situation when the subperiosteal hydrostatic transmission of pressures is obstructed (experiment 1). 

In the second setting (experiment 2), the plug was dissolved in the water after light manual manipulation (Figure 3(1)) so that the water inside the bag was evenly distributed everywhere, unimpeded.

To compare local loads with the obstructed versus unobstructed subperiosteal hydrostatic transmission of pressures, we used the DataLITE measuring system by Biometrics Ltd., Newport, UK. The system consisted of the Pinchmeter P200 with ±1.5% accuracy, a Ø = 7 mm measuring plate (Figure 4), and a DataLITE PIONEER Wireless Interface System with a USB connection to a computer. We embedded the pinch meter into a custom plaster mold so that its measuring plate was level with the surface of the plaster.

Experiment 1. The bag was placed on top of the pinch meter with the caramel plug positioned over the measuring plate (Figure 2(1A)). After zeroing the system to nullify the weight of the caramel plug and the bag, a Romeda 0.1 kg calibrated weight was placed on the bag over the caramel plug (Figure 2(2)). The measurement system registered a load on the plate of 0.1 kg, as shown in Figure 2(3).

Experiment 2. We wanted to see if the load on the plate would be reduced if the plug, which acted as an impediment to hydrostatic pressure transmission within the fluid film between the bag’s walls, was removed. When the plug was dissolved after light manual manipulation (Figure 3(1)) and the 0.1 kg weight was placed over the space above the measuring plate where the plug had been but which was now filled with water (Figure 3(2)), the measurement system registered a load on the plate of 0.07 kg, as shown in Figure 3(3), which was 30% less than in experiment 1.

## 3. Data Analysis

Each experiment was replicated 10 times. The data collection consisted of recordings on the pinch meter under two conditions. In experiment 1, the 0.1 kg weight was placed over the plug above the pinch meter. In experiment 2, the 0.1 kg weight was placed over the pinch meter after dissolving the caramel plug. The raw data with related statistics of the mean, standard deviation, *t*-score, and *p*-value are presented in Table 1.

We ran a paired (dependent sample) *t*-test. The pairs were formed naturally with the measurements of the obstructed (plugged) and unobstructed (unplugged) experimental conditions. Each rerun of the experiment, with a fresh bag and caramel plug, supplied the data measurements. The differences in the readings from the measuring plate pinch meter were recorded. The mean difference was 0.031 kg, with a standard deviation of 0.0032 kg. The null hypothesis was that the population average difference in paired weight measurements was equal to 0, against a two-sided alternative that it was not equal to 0. The test was performed at the 0.05 level of significance. The *t*-statistic was 9.8 with a corresponding *p*-value less than 0.000001. The null hypothesis was therefore rejected, with the conclusion that there was a strong statistically significant difference in weight measurement across the obstructed and unobstructed experimental conditions.

## 4. Results

When a weight of 0.1 kg was placed over the wall’s spot covering the caramel plug, the load on the measuring plate of the pinch meter (see experiment 1) was 0.1 kg each time.

When the same weight of 0.1 kg was put over the same wall’s spot after the caramel plug had been dissolved (see experiment 2), the load recorded on a measuring plate of the pinch meter was 0.07 kg each time, except for one measurement of 0.06 kg. The granularity of the measurement device was 0.01 kg. That there was almost no variability in the recordings for each experimental condition, respectively, was expected, as the same experimental setup was replicated for each run of the experiments.

The fact that the load on the measuring plate in experiment 2 was noticeably smaller than in experiment 1 indicates that local pressure is reduced when hydrostatic pressure transmission is not interrupted within the volume of the fluid film between the walls of a container. In the setting of both experiments, the flat bottom of the weight was positioned against the flat measuring plate (Figure 4) with the plastic bag between them. In experiment 1, the solid caramel plug effectively isolated the area under the weight from the water inside the bag (Figure 2(2)). By contrast, in experiment 2, a thin layer of water not only replaced the dissolved plug, but provided the continuous hydrostatic connectivity of the area under the weight with the entire water volume in the bag. It allowed the bag’s walls to assume a portion of the load from the weight on the measuring plate. This distribution was governed by Pascal’s Law.

## 5. Discussion

With certain limitations, this model can be interpreted as an additional objective argument in favor of the Floating Skeleton theory. The theory posits that joint capsules are hydrostatically connected, and therefore that an increase in pressure in one joint is transmitted between the capsules, with pressures redistributed through the periosteal sheath, and thus the load on the joint itself is reduced. But if there is a blockage and the hydrostatic connectivity is plugged, then the pressure applied to a given joint is not redistributed and reduced; persistently high, undistributed pressure has the potential to cause pathological musculoskeletal consequences.

As with any other physiological system in the human body, this Floating Skeleton system has multiple integrative features that support its contribution to the well-being of the higher organism and for its own sustainability and development. The investigation of these integrative features will be the subject of future multidisciplinary efforts. The simulation presented in this paper, which lowered the load on cartilage within the Floating Skeleton system, allows us to propose concrete ideas for how to prolong the duration of the protective role of the Floating Skeleton system in normalizing loads on cartilage. It has practical implications for public health, considering the inevitable thinning of the periosteum and subperiosteal space with age [23], which makes it more challenging to preserve the protective function of the Floating Skeleton system. The exigency of improving periosteal health is not only a long-term consideration: relaxed musculature during our sleeping hours does not stretch the periosteum as it does throughout the day, causing the natural shrinkage of the elastic tissues of the periosteum. Its tightening may result in the expulsion of the subperiosteal fluid necessary for the hydrostatic connectivity of one or many joints [24], leading to more frequent injuries when first stepping out of bed [25].

Muscle contraction mechanically activates subperiosteal impregnation by synovial fluid. Tendons and ligaments are attached to the bone partially via the periosteum at the entheses sites [26]. Muscle contraction stretches the periosteum in the entheses, which “pumps” extra fluid into the subperiosteal space [27]. This is the candidate natural mechanism for activating the pressure transmission between joints, which works automatically and unconsciously sustains the Floating Skeleton system’s normal functioning. The more entheses sites that are involved in such pumping in the pressure transmission route between joint capsules, the better the conditions for maintaining the subperiosteal fluidic conductivity. That brings our attention to the fibrous entheses providing periosteal–diaphyseal attachments which are found in the central part of the long bone shafts of long bone muscles, like the deltoid, which is inserted into the humerus, and the muscles attached to the linea aspera of the femur, such as the adductor magnus [26,28]. These are the muscles which participate less in the routine motions of body segments, but with critical contributions to challenging activities in competitive sports, martial arts, ballet, etc. These activities are longitudinal and frontal splits, the bending of the torso backward, forward and to the sides, and the twisting of the torso in the warrior stance. They all are components of well-known traditional methods of physical improvement like yoga, tai-chi, karate, and sumo, which notably endow their practitioners with a physical superiority over the regular public; practitioners who practice for many years can overcome possible related traumas, for example due to landing from high jumps.

The need to maintain uninterrupted subperiosteal hydrostatic conductivity suggests a fresh consideration of the existing systems of physical exercise. A starting point should be a review of the link between any given specific exercise and the location and type of entheses transmitting the tension of the muscles involved, for example, movements activating the muscles whose fibrous entheses pair the femur with the pelvis or the pelvis with the spine. The rationale is that the contraction of these muscles stimulates uninterrupted pressure transmission through the pelvis as a hub between the joint capsules of the left and right legs and between the legs and the spine.

One author developed the first rehabilitative method that focuses on maintaining the normal functioning of the Floating Skeleton system, called *Sanomechanics*^®^ [15,17]. The exercises of Sanomechanics are designed to create conditions for the subperiosteal space to be consistently impregnated with synovial fluid and for the Floating Skeleton system to remain unblocked. A mental component of Sanomechanics—autosuggestion—helps to visualize the Floating Skeleton system to properly tune the exercises.

The exercises belonging to this method are calibrated by a “criterion of correctness” according to the signals of encouragements the body generates [29], as postures and exercises generating pleasurable sensations are beneficial for restoring the hydrostatic connection of synovial capsules within the Floating Skeleton system (Figure 1b). As the existing methods dictate “what to do”, the Sanomechanics criterion of correctness can help in answering the questions of “why” to do this exercise and “how to do it”.

### Trigger Points: Additional Consideration

The periosteum is a part of the fascial system, which is also made up of fasciae covering the muscles (myofasciae) [30]. Together, a muscle and its fascia form the myofascial unit defined in [31]. It is logical to extend the model of pressure distribution via the periosteal sheath to the hydromechanics of the myofascial unit, with applications to myofascial pain syndromes [32].

Let us consider the phenomenon of trigger points in association with myofascial pain [33], like pain in the lower back [34], neck [35], and pelvis [36]. Ultrasound elastography can demonstrate an increased stiffness and change in structure in the muscle in the areas of trigger points [37], but specific mechanisms by which the trigger points develop and why they disappear after certain procedures are still unknown [38]. 

One of the methods for decreasing stiffness is the injection of normal saline into a specific fascial layer of the myofascial unit under ultrasound control once the location of the painful zone has been identified with palpation [39]. Another quite popular one is the myofascial relief method patented by Joseph Jacobs [40]. A meta-analysis of 40 studies showed that manual therapy, as well as laser therapy and extracorporeal shock wave therapy, can effectively reduce pain intensity, pressure pain threshold, and pain-related disability, with statistical significance when compared to the placebo [41].

Manual therapy within the myofascial relief method is in essence an instruction for manipulations, with pressures applied within the painful areas using fingers or devices. With some stylization, one may see the similarity of these manipulations to those demonstrated in Figure 3(1), where manipulations were used to more quickly dissolve the caramel plug and restore the fluid layer between the walls of the Ziploc bag. The closest anatomical similarity to our hydromechanical model is a painful area with minimal if any muscle bundles between the skin and the underlying bone. Manipulations there can easily reach and affect the periosteum sheath and, according to our theory, manipulations restore the layer of subperiosteal fluid, thus lowering the pressure applied by the masseur to the bone. And pain relief is nothing other than making the applied pressure smaller than the pain threshold of the bony surface.

When manipulations affect the myofascial units between the skin and the bone, they stimulate the local delivery of interstitial fluid, which facilitates the distribution of pressure by the fascia.

## 6. Conclusions

The experiments presented in this paper modeled:
(a)Loading between two surfaces without a layer of fluid between them (experiment 1) and with a layer of fluid between them (experiment 2).A documented reduction in loading in experiment 2 may help to provide a better understanding of the effect of recently discovered pressure sharing between the joint capsules (Floating Skeleton system) protecting contacting cartilages.Future studies may suggest that methods focused on maintaining/restoring the normal functioning of the Floating Skeleton system can be harnessed by more efficient rehabilitative strategies.

## Figures and Tables

**Figure 1 biomimetics-09-00222-f001:**
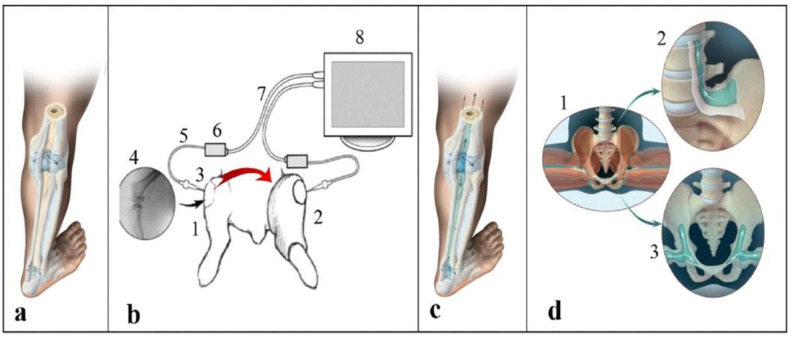
(**a**–**c**) Previous experiments demonstrating the subperiosteal transmission of pressures between joint capsules [9]: (**a**) traditional understanding of hydrostatic isolation of the synovial capsules from each other; (**b**) simultaneous pressure measurements in a movable right knee joint (1) and an immobilized left knee joint (2), confirmed subperiosteal hydrostatic transmission of pressures between the joints (curved arrow); cannulas (3) inserted laterally into the joint capsules (4) and connected via a fluid line (5) to the pressure sensor (6); the signal from the sensor (6) is transmitted via a cable (7) to the monitor (8); (**c**) artistic illustration of pressure transmission between joints via the subperiosteal space; (**d**) pull-out effect of contracting muscles on the periosteum in the entheses zones: in the split or the wide squat position (1), facilitating the hydrostatic transmission of pressures from the legs to the spine (2), and between the legs (3). Adopted from [12]. Video recordings of the animal experiments (**b**) are available as Appendix A.

**Figure 2 biomimetics-09-00222-f002:**
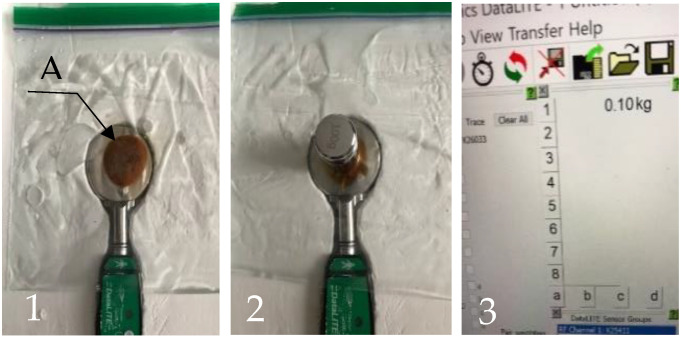
Effect on pressure transmission via a solid plug isolating the walls of the bag from the surrounding water. (**1**) Bag and the plug (A), made of caramelized sugar with water surrounding it; (**2**) placement of a weight of 0.1 kg above the plug; (**3**) load of 0.1 kg registered on computer screen.

**Figure 3 biomimetics-09-00222-f003:**
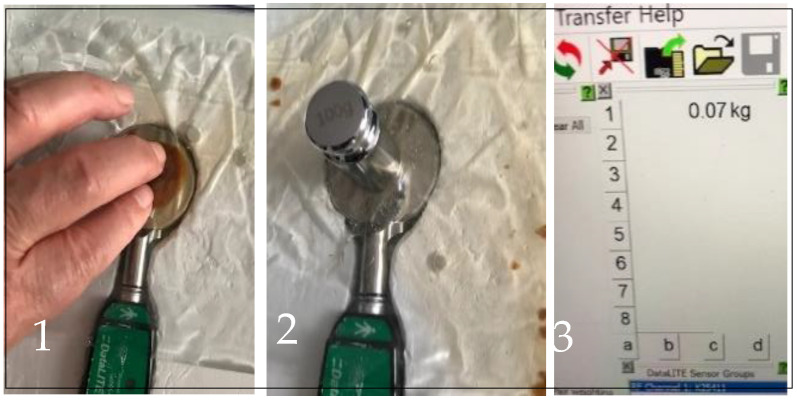
Effect on pressure transmission via restored layer of fluid. (**1**) Manipulations for breaking and dissolving the caramel plug; (**2**) the caramel plug is dissolved and the 0.1 kg weight is positioned over the bag; (**3**) load of 0.07 kg registered by the system on computer screen.

**Figure 4 biomimetics-09-00222-f004:**
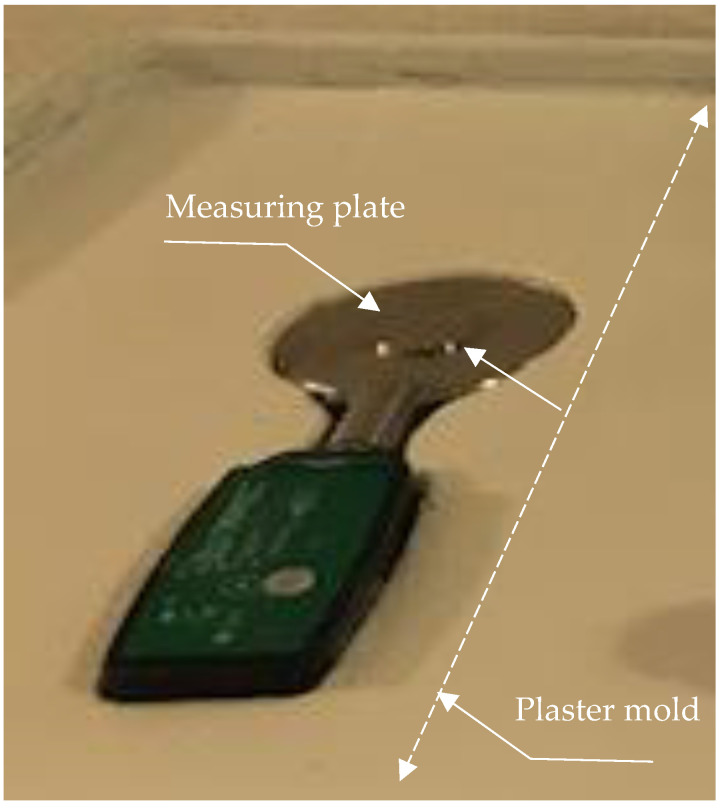
Pinchmeter P200 embedded in a plaster mold.

**Table 1 biomimetics-09-00222-t001:** The raw data and statistics of the mean, standard deviation, *t*-score, and *p*-value.

Experimental Run	Condition 1 (with Obstruction)	Condition 2 (without Obstruction)	Difference
1	0.1	0.7	
2	0.1	0.7	
3	0.1	0.7	
4	0.1	0.7	
5	0.1	0.7	
6	0.1	0.7	
7	0.1	0.6	
8	0.1	0.7	
9	0.1	0.7	
10	0.1	0.7	
Average	0.1	0.969	0.031
SD	0.0	0.0032	0.0032
*t*-statistics			9.8
*p*-value			<0.000001

## Data Availability

The data of this study on reasonable request are available from the author.

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
