# Peer review of "Modeling of the Effect of Subperiosteal Hydrostatic Pressure Conductivity between Joints on Decreasing Contact Loads on Cartilage and of the Effect of Myofascial Relief in Treating Trigger Points: The Floating Skeleton Theory"

_biomimetics, 2024, doi:10.3390/biomimetics9040222_

Round 1

Reviewer 1 Report

Comments and Suggestions for Authors

Dear Author, Please find enclosed the review report.

Comments on the Quality of English Language

Moderate editing of English language required

Author Response

Dear Reviewers, I am grateful for your professional comments, which helped guide revisions to the manuscript. Below are responses to specific comments and recommendations.

Reviewer 1

Thank you very much for your review. Your recommendations have been addressed in the revised manuscript with appreciation.

1.1. The title of the article is very long; it should be revised. the title of the article should be specific as per the context of the paper.

The title is quite long indeed, but it has been selected with purpose. The reason is that modeling presented relates to a newly discovered physical system which needs to be spelled out from the first lines.

1.2-1.3. The overall formatting of the paper is well below standard for instance the italic font has been used in the paragraph of the introduction which is not consistent.

As recommended, all italics were removed.

1.4. The conclusion of the paper is also not well written and it needs to be revised for instance reference has 7 been used inside the conclusion which is against the common method of general manuscripts writing formats. Example is [pressure sharing [7] that protecting contacting cartilages]. It should be revised and the conclusion should be enumerated with the sequence of the manuscript findings.

This reference has been removed from the Conclusion and this section has been modified as recommended.

1.5. As per biphasic analysis a strong correlations exist between the Ratio of interstitial fluid pressure to applied stress, this type of experimental or numerical outcomes are completely missing in the manuscript. for example the date showing the some preliminary graphs showing this phenomenon will add merit to the publications. For reference please review the findings presented in the following paper https://www.ncbi.nlm.nih.gov/pmc/articles/PMC2758165/

Thank you very much for these recommendations. Accordingly, references for publications 5 and 6 (PMC2758165) were made in the Introduction.

Reviewer 2

.

Thank you very much for your review. Your recommendations have been addressed in the revised manuscript with appreciation.

2.1. Laplace's law is a relationship between pressure and the surface tension. The statement cited by the authors is incomplete. It should read " Pressure inside an inflated elastic container with a curved surface is inversely proportional to the radius as long as the surface tension is presumed to change little." It is very important to mention the presence of surface tension. Also note that in the current experiment, we dont expect surface tension since the bag is expected to be filled completely. The effect of this law on the bag shape is not explained sufficiently

I am thankful for this highly professional comment. The corresponding citation has been corrected. The reference to Laplace’s law for the current experiments makes sense to the explanation of the effect of articulating of an anatomical joint on the inside pressure. That has been corrected in the Introduction section.

2.2. The paper contains more background information and research, while the experimental design and explanation are not very significant.

More details are given now for experimental design and the interpretation of the results.

2.3. When a weight is placed on a solid plug on a measuring scale, it is expected that the entire weight is read by the scale. If the plug is removed, the measured weight reduces since the bag might transmit some of the pressure to areas outside the measuring area of the scale, like the housing. The results of the experiment are not very scientifically significant, since the basic application of Pascal's law is well known. More investigation is required to support a statement such as "pressure has been reduced when hydrostatic pressure transmission is not interrupted within the volume of the fluid film between the walls of a container". 

The statement that "pressure has been reduced when hydrostatic pressure transmission is not interrupted within the volume of the fluid film between the walls of a container" is correct. This is because, according to Pascal’s Law, measured weight should be and is reduced since the bag might transmit some of the pressure to areas outside the measuring area of the scale.

2.4. The weighing scale measures total load present on the scale (which should be equal to the weight placed on the scale if the setup is isolated correctly). If the authors want to check the pressure in the bag, a pressure sensor should be used instead.

Both experiments are set up for measuring weight placed on the scale, not the pressure inside the bag.

2.5. There are many spelling mistakes and grammatical errors. They need to be fixed.

That is unfortunate indeed, and the revised text has been thoroughly edited.

Reviewer 3

Thank you very much for your review. Your recommendations have been addressed in the revised manuscript with appreciation.

3.1. The abstract is concise but could be more informative. It should briefly mention the key findings and their implications, providing a clear snapshot of the article's contribution.

The Introduction has been revised with this recommendation addressed.

3.2. The "Data analysis" section didn't present sufficient data for analysis. The author is highly recommended to present the 10 times' collected data in an intuitive format, such as line graph or table. 

The individual measurements are now presented in tabular form, along with summary statistics (mean, sd, t-statistic, and p-value).

3.3. The article mentions a paired t-test, but it would be beneficial to provide more details on the statistical analysis, including assumptions made, significance levels, and how p-values were interpreted.

Thank you for this recommendation. Additional detail has been incorporated into the discussion of the statistical analysis.

3.4.The conclusion could be more explicit in summarizing the key findings and their implications for the proposed theories. It should also suggest potential avenues for future research based on the current findings.

Thank you. Conclusion has been modified as recommended.

Reviewer 2 Report

Comments and Suggestions for Authors

The experimental method presented, is not accurate and this paper needs significant improvements. The following are the comments to the authors.

1) Laplace's law is a relationship between pressure and the surface tension. The statement cited by the authors is incomplete. It should read " Pressure inside an inflated elastic container with a curved surface is inversely proportional to the radius as long as the surface tension is presumed to change little." It is very important to mention the presence of surface tension. Also note that in the current experiment, we dont expect surface tension since the bag is expected to be filled completely. The effect of this law on the bag shape is not explained sufficiently.

2) The paper contains more background information and research, while the experimental design and explanation are not very significant. 

3) When a weight is placed on a solid plug on a measuring scale, it is expected that the entire weight is read by the scale. If the plug is removed, the measured weight reduces since the bag might transmit some of the pressure to areas outside the measuring area of the scale, like the housing. The results of the experiment are not very scientifically significant, since the basic application of Pascal's law is well known. More investigation is required to support a statement such as "pressure has been reduced when hydrostatic pressure transmission is not interrupted within the volume of the fluid film between the walls of a container". 

4)The weighing scale measures total load present on the scale (which should be equal to the weight placed on the scale if the setup is isolated correctly). If the authors want to check the pressure in the bag, a pressure sensor should be used instead.

Comments on the Quality of English Language

There are many spelling mistakes and grammatical errors. They need to be fixed.

Author Response

(The authors gave the same response as above.)

Reviewer 3 Report

Comments and Suggestions for Authors

The article presents an intriguing hypothesis regarding the Floating Skeleton theory and its potential implications for understanding joint health, cartilage loading, and myofascial relief. While the overall content is comprehensive, there are a few points to consider for improvement:

1. The abstract is concise but could be more informative. It should briefly mention the key findings and their implications, providing a clear snapshot of the article's contribution.

2. The "Data analysis" section didn't present sufficient data for analysis. The author is highly recommended to present the 10 times' collected data in an intuitive format, such as line graph or table. 

3. The article mentions a paired t-test, but it would be beneficial to provide more details on the statistical analysis, including assumptions made, significance levels, and how p-values were interpreted.

4. The conclusion could be more explicit in summarizing the key findings and their implications for the proposed theories. It should also suggest potential avenues for future research based on the current findings.

5. The references shall be consistent in format. Some references include URI or DOI links, while others do not. Please modify them accordingly to ensure format consistency. In addition, the U.S. patent (not Google Patent) referenced in [35] should include the formal Patent No. (US9649244B1) for the audience to refer to.

Comments on the Quality of English Language

The article presents an intriguing hypothesis regarding the Floating Skeleton theory and its potential implications for understanding joint health, cartilage loading, and myofascial relief. While the overall content is comprehensive, there are a few points to consider for improvement:

1. The abstract is concise but could be more informative. It should briefly mention the key findings and their implications, providing a clear snapshot of the article's contribution.

2. The "Data analysis" section didn't present sufficient data for analysis. The author is highly recommended to present the 10 times' collected data in an intuitive format, such as line graph or table. 

3. The article mentions a paired t-test, but it would be beneficial to provide more details on the statistical analysis, including assumptions made, significance levels, and how p-values were interpreted.

4. The conclusion could be more explicit in summarizing the key findings and their implications for the proposed theories. It should also suggest potential avenues for future research based on the current findings.

5. The references shall be consistent in format. Some references include URI or DOI links, while others do not. Please modify them accordingly to ensure format consistency. In addition, the U.S. patent (not Google Patent) referenced in [35] should include the formal Patent No. (US9649244B1) for the audience to refer to.

Author Response

(The authors gave the same response as above.)

Round 2

Reviewer 1 Report

Comments and Suggestions for Authors

Dear authors, thank you for the revision. I recommend paper for acceptance

Comments on the Quality of English Language

Minor English editing can be done further